# Valorization of Common (*Fagopyrum esculentum* Moench.) and Tartary (*Fagopyrum tataricum* Gaertn.) Buckwheat in Gluten-Free Polenta Samples: Chemical-Physical and Sensory Characterization

**DOI:** 10.3390/foods11213442

**Published:** 2022-10-30

**Authors:** Noemi Sofia Rabitti, Marta Appiani, Alessandra Marti, Susanna Buratti, Simona Benedetti, Giulia Chiodaroli, Cristina Proserpio, Monica Laureati

**Affiliations:** 1Sensory & Consumer Science Lab (SCS_Lab), Department of Food, Environmental and Nutritional Sciences (DeFENS), University of Milan, 20133 Milan, Italy; 2Department of Food, Environmental and Nutritional Sciences (DeFENS), University of Milan, 20133 Milan, Italy

**Keywords:** gluten-free, pseudocereals, polenta, electronic tongue, texture, color, sensory profile

## Abstract

In recent years, increasing attention has been addressed to buckwheat, an interesting minor crop with an excellent nutritional profile that contributes to the sustainability and biodiversity of the agri-food system. However, the high content of rutin and quercitin present in this pseudocereal can elicit undesirable sensory properties, such as bitterness and astringency, that can limit its exploitation in food formulations. The aim of the present study was to characterize six gluten-free porridge-type formulations (called polenta) prepared using corn and buckwheat flour. Specifically, polenta samples were prepared adding common (CB) or Tartary buckwheat (TB) flour at 20% (CB20; TB20), 30% (CB30; TB30), and 40% (CB40; TB40) to corn flour. Product characterization included sensory and instrumental analyses (electronic tongue, colorimeter, and Texture Analyzer). Products containing Tartary buckwheat were darker, firmer, and characterized by a higher intensity of bitter taste and astringency than those prepared with common buckwheat. In this context, the impact of buckwheat species seems to be more important at 30% and 40% levels, suggesting that lower additions may mask the differences between the species. The gathered information could support the food industry in re-formulating products with buckwheat. Finally, findings about the relationship between instrumental and sensory data might be exploited by the food industry to decide/choose what indices to use to characterize new formulations and/or new products.

## 1. Introduction

In recent years, a great deal of interest has been directed toward the topic of food sustainability to achieve a transformation along the entire food chain aimed at improving food systems [1]. To ensure this change, an agricultural production that conserves biodiversity, provides essential nutrients for a balanced diet, and increases the volume of minor crops is needed [2].

Tartary buckwheat (*Fagopyrum tataricum* Gaertn.) is a minor crop present in Europe, especially Slovenia, Northern Europe, and Italy [3]. It is a pseudocereal belonging to the Polygonaceae family, which is considered a sustainable crop because its growth is guaranteed even in the presence of unfavorable environmental conditions (e.g., low temperature, dry climate, and nutrient-poor soil [4,5]) and has a reduced need for fertilizers and pesticides [3].

This minor crop has interesting food applications; being gluten-free, it represents a potential ingredient for new food formulations targeted to people with celiac disease [6]. It also has an excellent nutritional profile, being rich in beneficial components for one’s diet [7]. For instance, buckwheat flour has a more balanced amino acid composition and thus a higher biological value than most cereals [8], as well as a good fatty acid composition (2%) consisting mainly of unsaturated (74.5%) versus saturated (25.3%) fatty acids [9]. Compared to other cereals, Tartary buckwheat contains a higher amount of soluble fiber (approximately 20–40% of the total fiber; [10]), and it is characterized by a high amount of minerals (K, Mg, Ca, Na, Cu, Zn, Fe, and Mn) and B vitamins [8,10]. This pseudocereal also has a high content of total flavonoids [11], mainly represented by rutin and quercetin, which are significantly higher in Tartary buckwheat than in common buckwheat (rutin: 16—up to 1418 and 7—up to 168 mg/100 g for Tartary and common buckwheat, respectively; quercetin: 31—up to 857 mg/100 g and up to 9 mg/100 g, for Tartary and common buckwheat, respectively; [11]). Therefore, the incorporation of buckwheat into foods enhances its bioactive potential by increasing the levels of these two flavonoids, particularly rutin [12,13].

While representing a biological advantage due to their antioxidant [14] and anti-inflammatory [15] activity, rutin and quercetin are responsible for the bitterness and astringency perception of buckwheat-based products [16,17], which may negatively influence consumer preferences [18,19]. Moreover, the addition of common buckwheat is suitable only in certain types of foods. Common buckwheat milk, for example, besides being bitter, does not have a stable hydrocolloid system, thus affecting its rheological and sensory properties [20].

Until now, common buckwheat has been added to different types of foods, such as bread [12], pasta [13,21], extruded snacks [22], and plant-based milk [20], to enhance their nutritional profiles. However, very few studies have been performed on Tartary buckwheat. In a recent study, Buratti et al. [16] showed that the pre-gelatinization process (i.e., cooking the flour in an excess of water) decreased the rutin content, with a positive effect on bitterness and astringency reduction in both common and Tartary buckwheat. Hitherto, the effect of common and/or Tartary buckwheat addition on taste perception has been explored only in raw materials (e.g., flour) using electronic devices such as an electronic nose (e-nose) and electronic tongue (e-tongue) [16], but there are no studies that consider food formulations and combine sensory and instrumental analyses. To fill this gap and to obtain a more comprehensive description of Tartary buckwheat-enriched food products, the aim of the present study was to characterize six gluten-free porridge-type products (called polenta) based on corn and buckwheat flour through a sensory analysis conducted by a trained panel of assessors, and instrumental analyses including e-tongue, colorimeter, and texture analyzer. The relationship between sensory properties assessed by human senses and instrumental devices was also investigated.

## 2. Materials and Methods

### 2.1. Samples

Flours from common (CB) and Tartary buckwheat (TB) were purchased by Raetia Biodiversità Alpine (Teglio, Sondrio, Italy), whereas corn flour was kindly provided by Molino Filippini S.r.l. (Teglio, Sondrio, Italy). CB or TB flour was added at 20% (CB20; TB20), 30% (CB30; TB30), and 40% (CB40; TB40) to corn flour. Mixtures were cooked in the conditions reported in Appiani et al. [18] to prepare a porridge-type product, which is known as “polenta” in Northern Italy. In total, six formulations were considered.

Polenta formulations were prepared using an automatic cooking mixer (Thermomix TM 31—Vorwerk Contempora S.r.l., Milan, Italy) in which 400 g of mixture (corn flour with CB or TB flour) was added to 1.5 kg of water salted with 8 g of NaCl and brought to a boil. After boiling for 40 min, the polenta samples were brought to room temperature.

Samples were prepared the day before the evaluations and stored at 4 °C in aluminum trays (Figure 1).

On the day of evaluations, formulations were removed from the refrigerator two hours before the analysis.

### 2.2. Instrumental Analyses

#### 2.2.1. Electronic Tongue

E-tongue measurements were carried out using Taste-Sensing System SA 402B (Intelligent Sensor Technology, Inc. 5-1-1 Onna, Atsugi City, Kanagawa Prefecture 243-0032 Japan). The system consists of detecting sensors whose surface is attached to artificial lipid membranes with different response properties to chemical compounds on the basis of taste. In this work, four detecting sensors (CT0 for saltiness, AAE for umami, C00 for bitterness and aftertaste bitterness, and AE1 for astringency and aftertaste astringency) were applied, separated into two arrays according to membrane charge.

For e-tongue measurement, polenta samples were frozen at −20 °C, freeze-dried, and ground using a mortar and pestle, then prepared as reported by Buratti et al. [16]. E-tongue measuring process was previously reported by Laureati et al. [23].

Each sample was evaluated in triplicate, and the “taste values” were calculated by multiplying sensor outputs by appropriate coefficients based on Weber-Fechner law, which provides the intensity of sensation based on the sensor properties for taste [24].

#### 2.2.2. Colorimeter

The color of the products was measured using a reflectance color meter (CR 210, Minolta Co., Osaka, Japan) to establish the lightness and saturation of the color intensity. Results were expressed in the CIE L* a* b* color space, where L* is luminosity, a* is redness/greenness, and b* is yellow/blueness. The following indices were calculated and considered for the discussion: browning index (100-L*) and a*/b* ratio as a measurement of the yellow (<1) or the red color (>1). Each polenta sample was placed in two trays, and color measurements were taken in six points for each tray.

#### 2.2.3. Texture Analyzer

Texture analysis was assessed with a TA.XTplusC Texture Analyser (Stable Micro Systems, Godalming, UK) performing a compression test and a Texture Profile Analysis (TPA). For both tests, the samples were cut into twelve discs with a diameter of 45 mm: six discs were used for the compression test and six discs for the TPA.

The compression test consisted of the compression of the polenta discs with a 36 mm Radiused Cylinder Probe until 25% strain with a penetration speed of 1 mm/s, followed by 60 s, during which the probe was held at the reached height. The parameters obtained were firmness and springiness. The former is the force (N) required to compress the sample starting from a 5 g trigger force. The latter is the ratio between the force after 60 s and the maximum force expressed in percentage.

TPA was performed with a 100 mm Compression Plate as a probe. The test consisted of a double compression of the sample discs with a 5 s pause in between. Each compression occurred at a speed of 5 mm/s until a 5 g force was opposed by the sample. The texture parameters considered were (a) Hardness (N), the maximum force obtained during the first compression, (b) Springiness, the ratio between the distance of the second compression peak and the distance of the first compression peak, (c) Cohesiveness, the ratio between the area under the second compression and the area under the first compression, (d) Gumminess, the product of hardness and cohesiveness, (e) Chewiness, the product of gumminess and springiness, and (f) Resilience, the ratio between the energy of the first compression and the energy of the second compression.

### 2.3. Sensory Analysis

#### 2.3.1. Participants

Sensory analysis was performed at the Sensory & Consumer Science Laboratory (SCS_Lab) of the University of Milan (Italy), designed in accordance with ISO guidelines [25]. The sensory profiling method [26] was applied to describe the sensory properties of the six polenta formulations. Eight subjects (8 women aged between 22 and 44 years) were selected among students and employees of the Faculty of Agriculture and Food Sciences of the University of Milan. Only subjects who were ≥18 years of age, who did not suffer from food allergies or intolerances, and who liked food formulations based on corn and buckwheat flour were involved in the study.

The present study was performed according to the principles established by the Helsinki Declaration, and the protocol was approved by the Ethical Committee of the University of Milan (n. 11/21).

#### 2.3.2. Experimental Procedure

Subjects were trained over a period of one month. During the training period, judges were informed about the aim of the study and familiarized with both method and food samples. Firstly, judges were trained to identify and rank basic tastes (sweetness, sourness, and bitterness) and astringency in aqueous solutions. Subsequently, judges were presented with the six samples to develop a common vocabulary for describing sensory attributes. This phase was performed in a quiet room designed for collective sessions. Each descriptive term was extensively defined to avoid any doubts about its meaning. Reference standards corresponding to the maximum intensity of the rating scale were developed to calibrate the assessors. The final list of the 6 attributes, definitions, and reference standards used for the final sample evaluation is reported in Appendix A.

Once common vocabulary and reference standards were developed, participants performed three preliminary sessions in sensory booths to familiarize themselves with the scale.

After the training phase, judges evaluated the six polenta samples in two replicates. The two replicates were performed on the same day with a 3-h break between the sessions. All judges performed the evaluation in individual sensory booths under white light and were asked to rate the intensity of each sensory descriptor by assigning scores ranging from 1 (almost total absence of the sensation) to 9 (maximum intensity of the sensation, represented by the reference standards). Judges had to refrain from smoking, eating, and drinking (except water) for the hour before tasting. On the day of evaluations, for both the training and evaluation sessions, samples were removed from the refrigerator two hours before the tasting session and cut into 8 cm^3^ cubes of approximately 15 g each, which were served at room temperature (approx. 20 °C) in small cups coded with three-digit numbers for each food formulation. Between sample evaluations, assessors were instructed to rinse their mouths with mineral water for 15 s. Sample presentation order was systematically varied across judges and replicates [27].

### 2.4. Data Analysis

Sensory data were analyzed using Analysis of variance (ANOVA) considering Products (6), Judges (8), Replicates (2) and their second-order interactions as factors, and sensory attributes as dependent variables.

Instrumental data related to color and texture were analyzed using 1-way ANOVA considering Products (6) as factors.

When the ANOVA showed a significant effect (*p* ≤ 0.05), the Least Significant Difference (LSD) was used as multiple comparison test.

E-tongue data were scaled with columns and elaborated on using Principal Component Analysis (PCA). PCA was applied as an exploratory tool to uncover in a reduced space the data structure and the relationships between objects and variables [28].

To study the relationship between sensory results and technological properties of samples, Partial Least Square Regression (PLSR) analysis was performed. PLSR models both the X- and Y-matrices simultaneously to find the variables in X that best predict the variables in Y. Instrumental data were considered as the X matrix and sensory data as the Y matrix. A correlation loadings plot was used to find variables with less than 50% explained variance that were left out of the model [29]. PCA and PLSR analyses were conducted using the XLSTAT software package (Addinsoft 2021, New York, NY, USA).

## 3. Results and Discussion

### 3.1. Instrumental Analyses

#### 3.1.1. Electronic Tongue

E-tongue data collected on the six polenta samples were elaborated using PCA. In Figure 2, the score plot (a) and the loading plot (b) showing the data structure and the relationship between objects and variables in the plane defined by the first and second principal components (PC1 and PC2) (explained variance 75.34%) are reported.

Considering the score plot (Figure 2a), common (CB) and Tartary (TB) samples were discriminated for their taste mainly along the PC1. In contrast, along PC2 samples were discriminated according to the percentage of buckwheat flour added to corn flour (20%, 30%, 40%). In particular, TB samples located at the right of PC1 were mainly perceived as more bitter and astringent than CB samples (Figure 2b). This outcome is in line with previous results found on common and Tartary buckwheat flour, in which the latter was found to be higher in bitter taste and astringency [16]. The present results confirm that Tartary buckwheat may also lead to increased bitterness when added to a food matrix. Furthermore, by increasing the percentage of buckwheat flour from 20% to 40%, samples were perceived as more and more characterized by umami and salty tastes. The positioning of salty and umami sensors in correspondence with Tartary buckwheat samples may be explained by the higher amount of minerals present in this pseudocereal [10].

#### 3.1.2. Color

Color results are reported in Table 1. The browning index (100-L*) increased significantly with the increased addition of both CB and TB. The increase in browning is mainly due to the nature of buckwheat flour, which is rich in fiber and ash [7,30]. At the lowest substitution level (20%), a significant difference was observed between CB and TB, where the samples containing CB showed the lowest browning. At higher substitution levels (i.e., 30% and 40%), no significant differences were observed based on the type of raw material.

By investigating the effect of buckwheat addition (from 10 to 30%) on corn extrudates color, Singh et al. [31] highlighted an increase in L* values (i.e., decrease in browning) up to the 10% buckwheat flour incorporation level and a decrease at the 20% level, followed again by an increase at the 30% level. Differences among the studies might be due to differences in processing conditions (i.e., cooking vs. extrusion). On the other hand, when buckwheat was used in corn-based bread, the rising buckwheat level of up to 40% in the formula caused a significant decrease in lightness [32]. A similar outcome was obtained by Brites et al. [12], who found a decrease in L* values in bread added to with 30% and 45% wholegrain buckwheat compared to refined buckwheat.

The increase in the a*/b* ratio suggests a decrease in yellowness upon buckwheat addition (Table 1), likely due to the decrease in the amount of corn meal, which is characterized by high b* values [33]. As regards the effect of the type of buckwheat, a*/b* values were significantly lower in samples containing CB; such differences were mainly due to the fact that the CB samples presented lower a* values than TB samples, with no differences in b* values (data not shown).

Color changes between TB and CB samples might be due to the different refinement levels of the flours, but as both were commercial samples, this information was not available. Similar to our findings, the overall redness increased with an increase in the level of buckwheat incorporation in both corn extrudates [31] and gluten-free bread [32].

#### 3.1.3. Texture

The results of the texture analysis of the buckwheat-enriched products are reported in Table 2. By increasing the CB-enrichment level from 20% to 30%, the firmness and hardness of the product significantly increased, whereas no significant differences were observed at the highest CB level (e.g., 40%). Changes in textural properties upon buckwheat addition might be due to the high fiber content. Indeed, an increase in gel hardness was observed when bran was added to the corn formulation [34]. On the contrary, a decrease in such indices was observed in a wheat-based formulation as the level of buckwheat increased from 17% to 67% [35]. Differences between our study and that of Yilmaz et al. [35] are likely due to the differences between the types of flour used as the base of the formulation (corn versus wheat) and potential interaction with the buckwheat, as well as the origin of the buckwheat and/or the refinement level of the flour. Gumminess and chewiness were significantly higher in samples with 30% CB addition compared to samples with 20% CB addition. This result was expected because gumminess and chewiness values must be influenced by hardness. Therefore, it was thought that the factors affecting the hardness would also affect the gumminess and chewiness values of the CB samples. As to resilience, the interaction of CB with corn had a decreasing effect on the capability of the product to regain its original position, in accordance with a previous study on the buckwheat-wheat system [35]. No differences in either cohesiveness or springiness (the capability of the material to withstand prolonged stress) were detected.

The type of buckwheat (CB versus TB) seemed to affect the texture of the product. Indeed, in the case of TB samples, only the firmness was significantly different between TB20 and TB30 (Table 2). The only significant—although slight—difference between TB20 and TB30 was observed for cohesiveness.

Finally, differences between CB and TB were significant only at the 20% level for hardness, gumminess, resilience, firmness and springiness (the latter measured using the compression test). Specifically, TB20 was higher in firmness, hardness, and gumminess but lower in resilience and springiness compared to CB20 and CB40, whereas TB40 was lower in springiness (as measured using TPA) than CB40 (Table 2). When comparing the physicochemical properties of the starch of three common and three Tartary buckwheat varieties, Li et al. [36] found a similarity in gel textural properties, likely due to the high experimental error in the measurement of the textural parameters.

### 3.2. Sensory Analyses

Panel reliability was satisfactory, as evidenced by the high repeatability between replicates and the high reproducibility among assessors (See Appendix A).

Sensory attribute mean values by product are reported in Figure 3. All sensory descriptors discriminated the samples (all *p* ≤ 0.001). Ochre color, bitterness, astringency, gumminess, and firmness increased significantly with the addition of common and Tartary buckwheat. An increased intensity in color and bitter taste correlating with the increasing addition of common buckwheat has also been demonstrated in other foods, such as pasta, cookies, and bread [12,21,37,38]. For instance, the incorporation of 30% and 45% wholegrain buckwheat flour in bread samples led to an increase in browning resulting in a decrease in crumb appearance and color acceptance, as well as overall impression [12]. Moreover, in pasta samples, the addition of 10%, 20%, and 30% of wholegrain buckwheat flour resulted in a significant increase in brown color intensity as well as bitterness [21]. The present results are also in line with previous findings depicting increased firmness due to the addition of this pseudocereal to dough samples [21]. Perceived moisture in the mouth decreased with the addition of common and Tartary buckwheat, with similar mean values observed for the two types of flour at the same concentrations. This result is coherent with the increase in perceived firmness and gumminess in samples with a higher addition of common and Tartary buckwheat flour, as the quantity of water contained in a food material is known to modulate texture perception [39]. However, a study conducted on bread samples with a 40% concentration of common buckwheat reported higher moisture than those at 20% and 30% [38]. To the best of our knowledge, there are still no studies in the literature that have employed trained judges to describe the sensory characteristics of foods enriched with Tartary buckwheat; therefore, further research is needed to confirm the present findings.

Sensory results were partially in agreement with instrumental data. Color evaluated both sensorially and instrumentally increased with the increased amount of common and Tartary buckwheat. However, the differences between the two types of flour were more pronounced when evaluated by the trained panel. The higher addition of common and Tartary buckwheat led to an intense perception of bitterness and astringency, as identified by both the sensory profile and e-tongue. A lower agreement between sensory and instrumental measurements was observed for texture parameters. The trained panel highlighted an increase in texture with the addition of both common and Tartary buckwheat flour, and a less delineated pattern was seen with the instrumental measurements.

### 3.3. Relationship between Instrumental and Sensory Data

A partial least square regression (PLSR) was applied to study the relationship between sensory and instrumental data in more detail. PLSR was conducted on all variables, including sensory and instrumental data (e-tongue, color, TPA, and compression test). All variables with less than 50% explained variance, according to a correlation loading plot analysis, were left out of the model. This analysis led to the exclusion of “Resilience” acquired by TPA, “Springiness” obtained by compression test, as well as bitterness (taste and aftertaste), as detected by e-tongue.

The positioning of the samples is reported in the Scores plot (Figure 4a). In contrast, the relation between sensory data (purple font) and instrumental data (TPA yellow font, colorimeter blue font, e-tongue green font, compression test red font) is reported in the Loadings plot (Figure 4b).

The first factor explains, respectively, the 46% and 91% of the variation in Y, while the second factor accounts for, respectively, the 33% and 2%. 

The first factor differentiated samples according to the level of buckwheat flour addition. Looking at the second factor, CB20 is opposed to TB20. 

Comparing Figure 4a,b, TB30, TB40, and CB40 were mainly described as having a perceived bitter taste (detected by trained judges) and astringency detected both by trained judges and electronic tongue. The TB30, TB40, and CB40 samples were also described as having a more intense ochre color as well as higher values of the parameters browning (L*-100) and a*/b*, which suggest a decrease in yellowness, indicating a good agreement between the sensory and instrumental measurements. The high intensity of these three characteristics may be a driver of the rejection of these products by consumers. A previous study has shown that consumers’ acceptance decreases significantly with the increase of Tartary buckwheat in polenta samples due to too strong a dark color, bitterness, and overall flavor, although the addition of up to 30% of this pseudocereal was still feasible [18]. Moreover, the same study has identified large differences in the liking of Tartary buckwheat enriched polenta samples, with a non-negligible proportion of consumers (30%) preferring polenta samples with the highest addition (up to 40%) of this pseudocereal [18].

As regards texture dimension, sensory data revealed that TB30 and TB40 samples were described according to sensory firmness and gumminess, as well as springiness and cohesiveness, measured using TPA, whereas the firmness measured using the compression test, was mainly associated with TB20 and CB30 samples. This latter sample is also positioned close to TPA data about gumminess, hardness, and chewiness. It is well known that the effect on the texture of an ingredient is highly product-dependent. For example, in food matrices such as bread [32] and potato-based pasta (gnocchi) [40], a decrease in food hardness has been shown with the increasing amount of buckwheat flour. In contrast, higher hardness was observed in noodles samples with higher additions of this pseudocereal [41].

CB20 was the sample with the lowest intensity of texture attributes (sensory and TPA), bitterness (sensory), and astringency (sensory and e-tongue) and the highest moisture content and yellow color compared to the other samples.

## 4. Conclusions

The results obtained by the present study provided insights into the development of a gluten-free porridge-type food enriched in buckwheat. Firstly, the role of the type of buckwheat (common versus Tartary) and enrichment level (20%, 30%, and 40%) on product sensory and chemico-physical features was assessed. Tartary buckwheat samples were darker, firmer, and characterized by a higher intensity of bitter taste and astringency than common buckwheat samples. In this context, the impact of buckwheat species seems to be more important at the 30% and 40% levels, suggesting that lower additions may mask the differences between the species. The gathered information could support the food industry in re-formulating products with buckwheat. Finally, findings about the relationship between instrumental (e-tongue, TPA, and compression test) and sensory data might be exploited by the food industry to decide/choose what indices are used to characterize new formulations and/or new products that have nutritional benefits for the consumer.

## Figures and Tables

**Figure 1 foods-11-03442-f001:**
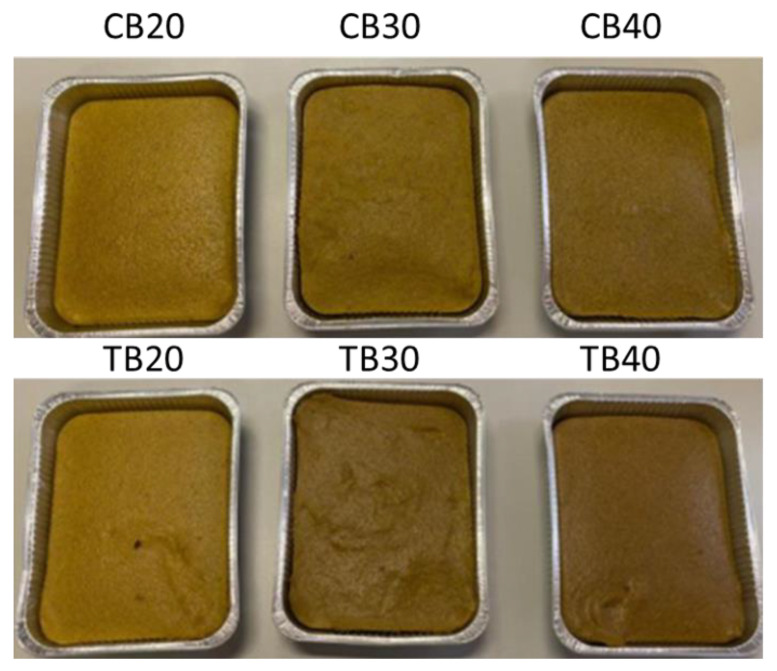
Polenta formulations with Common (CB) and Tartary buckwheat (TB) at 20% (CB20; TB20), 30% (CB30; TB30), and 40% (CB40; TB40).

**Figure 2 foods-11-03442-f002:**
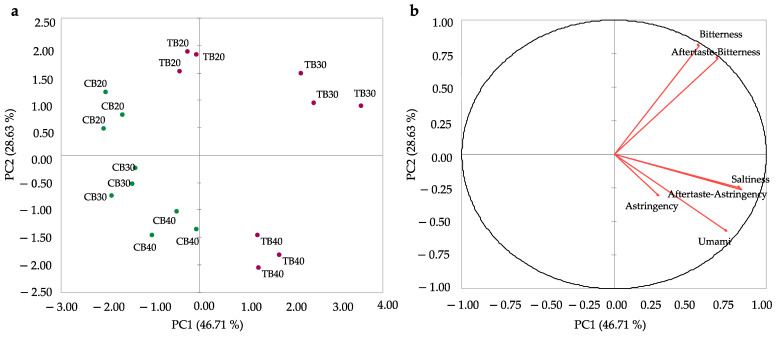
PCA score plot (**a**) and PCA loading plot (**b**) of e-tongue data collected on polenta enriched in common (CB) and Tartary (TB) buckwheat flour at various ratios (20%: CB20, TB20; 30%: CB30, TB30; 40%: CB40, TB40).

**Figure 3 foods-11-03442-f003:**
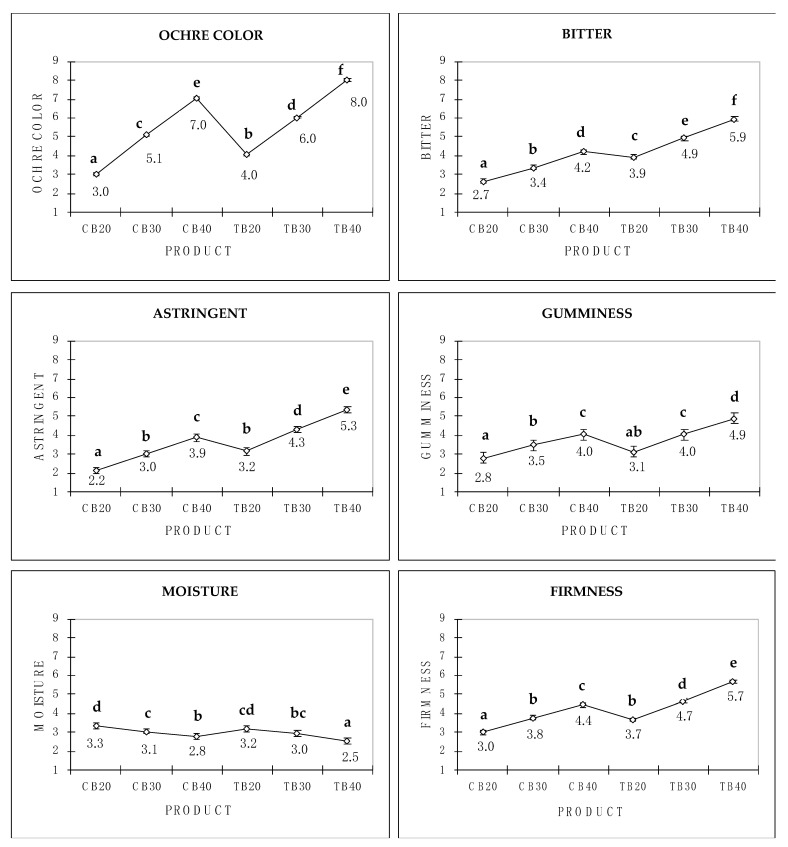
Sensory attribute mean values by product (CB = Common buckwheat; TB = Tartary buckwheat). Error bars represent standard error of means (SEM). Different letters indicate significant differences according to LSD post hoc test (*p* ≤ 0.05).

**Figure 4 foods-11-03442-f004:**
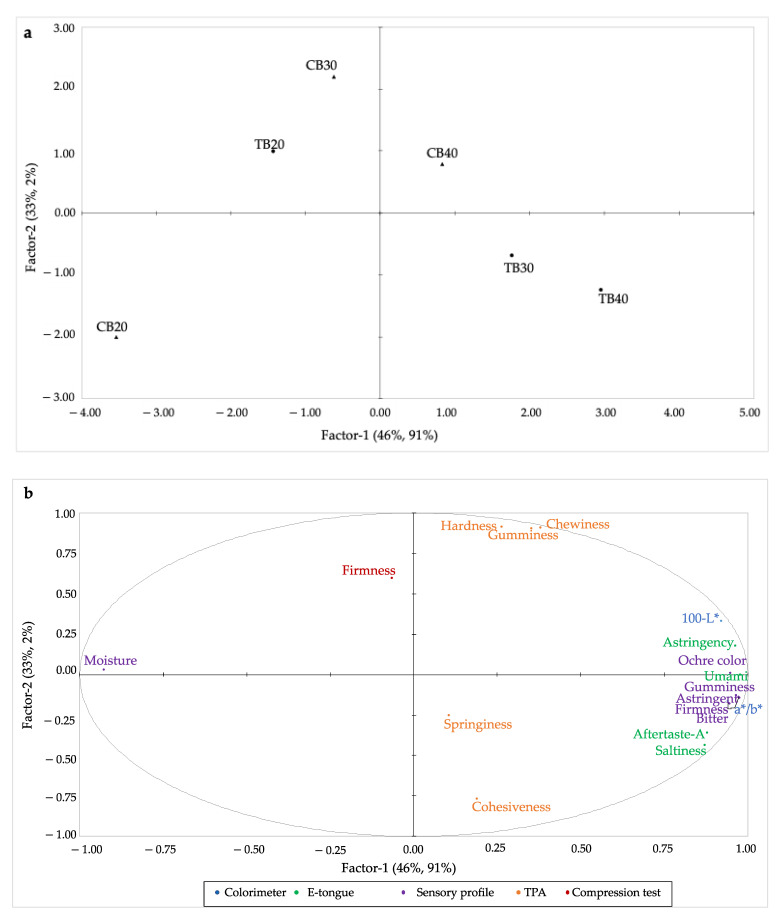
Score plot (**a**) and loading plot (**b**) obtained by the PLSR model of sensory analysis and instrumental determinations for each sample (CB20, CB30, CB40, TB20, TB30, TB40) (CB = common buckwheat; TB = Tartary buckwheat). Aftertaste-A = aftertaste-astringency.

**Table 1 foods-11-03442-t001:** Color attributes of polenta prepared from corn meal enriched in common (CB) and Tartary (TB) buckwheat flour at various ratios.

	100-L*	a*/b*
CB20	37.85 ± 0.58 a	0.11 ± 0.01 a
TB20	41.00 ± 1.15 b	0.17 ± 0.01 b
CB30	43.94 ± 1.11 c	0.16 ± 0.01 b
TB30	43.98 ± 0.84 c	0.22 ± 0.01 d
CB40	45.59 ± 1.12 d	0.20 ± 0.01 c
TB40	45.99 ± 0.36 d	0.31 ± 0.01 e

CB20, corn: CB = 80:20; CB30, corn: CB = 70:30; CB40, corn: CB = 60:40; TB20, corn: TB = 80:20; TB30, corn: TB = 70:30; TB40, corn: TB = 60:40. 100—L*, browning index; a*/b*, redness/yellowness ratio. Mean (*n* = 12) ± standard deviation followed by different letters in the same column indicates significant differences (LSD test; *p* ≤ 0.05).

**Table 2 foods-11-03442-t002:** Instrumental texture analysis of polenta samples prepared from corn meal enriched in common (CB) and Tartary (TB) buckwheat flour at various ratios.

	Texture Profile Analysis						Compression Test	
	Hardness (N)	Springiness	Cohesiveness	Gumminess	Chewiness	Resilience	Firmness (N)	Springiness (%)
CB20	45.19 ± 2.59 a	0.91 ± 0.02 ab	0.77 ± 0.01 ab	34.56 ± 1.94 a	31.44 ± 1.83 a	0.52 ± 0.01 b	28.45 ± 3.61 a	62.58 ± 2.39 b
TB20	66.34 ± 16.48 b	0.91 ± 0.01 ab	0.74 ± 0.03 a	47.70 ± 10.90 b	42.52 ± 9.93 ab	0.47 ± 0.04 a	42.05 ± 10.20 c	60.44 ± 2.45 a
CB30	64.80 ± 15.38 b	0.91 ± 0.01 ab	0.75 ± 0.04 ab	48.01 ± 9.32 b	43.31 ± 8.17 b	0.48 ± 0.04 a	39.58 ± 1.73 bc	63.50 ± 0.89 b
TB30	55.77 ± 18.77 ab	0.91 ± 0.01 ab	0.77 ± 0.03 b	42.51 ± 13.00 ab	38.95 ± 12.13 ab	0.50 ± 0.04 ab	33.10 ± 8.55 ab	62.06 ± 1.29 ab
CB40	60.87 ± 3.28 ab	0.92 ± 0.01 b	0.75 ± 0.01 ab	46.58 ± 0.61 b	42.70 ± 0.82 ab	0.48 ± 0.01 a	30.24 ± 5.23 a	63.93 ± 0.98 b
TB40	59.00 ± 8.92 ab	0.90 ± 0.03 a	0.75 ± 0.01 ab	44.52 ± 5.69 ab	39.92 ± 5.82 ab	0.47 ± 0.02 a	32.068 ± 3.54 ab	60.60 ± 1.13 a

CB20, corn: CB = 80:20; CB30, corn: CB = 70:30; CB40, corn: CB = 60:40; TB20, corn: TB = 80:20; TB30, corn: TB = 70:30; TB40, corn: TB = 60:40. Mean (*n* = 6) ± standard deviation followed by different letters in the same column indicates significant differences (LSD test; *p* < 0.05).

## Data Availability

Data will be made available upon reasonable request to the corresponding author.

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
