# Peer review of "Valorization of Common (Fagopyrum esculentum Moench.) and Tartary (Fagopyrum tataricum Gaertn.) Buckwheat in Gluten-Free Polenta Samples: Chemical-Physical and Sensory Characterization"

_foods, 2022, doi:10.3390/foods11213442_

Round 1
Reviewer 1 Report
It was very difficult to identify the product that was studied. My suggestion is kindly name the product and bring a clarity to your work. Throughout the manuscript no where the product name has been given except for one line in materials and methods. The kind of product has not been specified was is ready to eat or ready to cook or just a premix for preparing the product. What was studied is difficult to find out. After identifying the product, I believe The design of experiments is good and have been well presented.
After identifying the product which I believe the product was polenta which itself is a delicacy made with buckwheat and corn which is a porridge. The title or the abstract or the keyword should contain the product and/or the type of product for larger visibility in search.
The effect of various level of common buckwheat and tartary buckwheat was studied. The title, abstract, introduction and material and methods must be improved.
Results have been given and analysed well. The discussion can be improved with recent works and their results. Discussion must be improved for validating the finding of present work
Reviewer 2 Report
Dear Authors,
your paper has been sent for my consideration. In it, you investigate the common and tatary buckwheat as an ingredient in gluten-free food formulations. The relationships between the properties of the obtained product and its sensory acceptance were analyzed.
I must admit that I haven't read such an understandable, concise, and well-written article for a long time, which I highly praise. I also appreciate correct references.
I only have a few minor notes that I added in the PDF file.

Reviewer 3 Report
ArtykuÅ‚ jest dobrze napisany i skoncentrowany oraz zgodny z celami i zakresem czasopisma. JÄ™zyk angielski jest odpowiedni i zrozumiaÅ‚y, wymagane jest tylko niewielkie sprawdzanie pisowni. Ponadto metody sÄ… opisane z wystarczajÄ…cymi szczegóÅ‚ami, aby umożliwić innemu badaczowi odtworzenie protokoÅ‚u. SiÅ‚a naukowa artykuÅ‚u jest bardzo szczegóÅ‚owa i dobrze przeprowadzona i dobrze opisana. Po kilku drobnych poprawkach warto jÄ… opublikować. RÄ™kopis porusza interesujÄ…cy temat, a przed publikacjÄ… sÄ… drobne poprawki. Wymagane.
Tytuł
może warto umieścić łacińską nazwę gryka tartarna (Fagopyrum tataricum Gaertn).
Abstrakt
zbyt maÅ‚o informacji na temat zastosowanej metodologii i uzyskanych wyników
ProszÄ™ szczegóÅ‚owo opisać analizowane próbki i wskazać, że byÅ‚a to polenta
Materiały i metody
Próbki
nigdzie nie znaleźć żadnych informacji. że produkt jest polenta, pojawia się dopiero w dalszej części tekstu, proszę wypełnić
l.77 zsumować w miejscu zakupu mieszanin
l.82 w jakiej temperaturze jest gotowane
Czy w 1992 roku próby powtórzyÅ‚y siÄ™ dwukrotnie?
Wnioski
392 dlaczego jest to ważne? napisz frazę o właściwościach odżywczych gryki
Reviewer 4 Report
This manuscript describes a comparison study between different food formulations using common and Tartary buckwheat. Numerous tests were performed to characterize the six gluten-free food formulations based on corn and buckwheat flour. However, this study requires many additional experiments to strengthen the article, including its phytochemical contents, antioxidant properties, and biological activities.
The abstract section should be contained the background of the study.
The introduction section should be ameliorated by adding the newest evidence on common and Tartary buckwheat.
In the methods section: Authors are invited to describe the six gluten-free formulations studied
Round 2
Reviewer 1 Report
The changes have made the work very understandable and easy for readers and academicians.
The manuscript has bee well improved
Author Response
The changes have made the work very understandable and easy for readers and academicians.
Dear reviewer, we thank you for your positive comment.
The manuscript has been well improved.
We are glad that our manuscript has been improved as suggested.
Reviewer 4 Report
The authors have applied suggestions and they have responded to my questions. In this state, the manuscript is suitable for publication.
Author Response
The authors have applied suggestions and they have responded to my questions. In this state, the manuscript is suitable for publication.
Dear reviewer, we thank you again for your suggestions. We are pleased that now our manuscript is suitable for publication.